The tetrapod fauna of the upper Permian Naobaogou Formation of China: 1. Shiguaignathus wangi gen. et sp. nov., the first akidnognathid therocephalian from China

Liu Jun liujun@ivpp.ac.cn 1 2
Abdala Fernando Nestor.Abdala@wits.ac.za 3 4
1 Key Laboratory of Vertebrate Evolution and Human Origins of Chinese Academy of Sciences, Institute of Vertebrate Paleontology and Paleoanthropology, Chinese Academy of Sciences , Beijing , China
2 University of Chinese Academy of Sciences , Beijing , China
3 Evolutionary Studies Institute, University of the Witwatersrand , Johannesburg , South Africa
4 Unidad Ejecutora Lillo (CONICET-Fundación Miguel Lillo) , Tucumán , Argentina
Sues Hans-Dieter
Electronic publication date: 2017 Dec 6
Publication date: 2017
Volume: 5
Electronic Location ID: e4150
Received 2017 Sep 20; Accepted 2017 Nov 18
Copyright: ©2017 Liu and Abdala
Copyright year: 2017
Copyright holder: Liu and Abdala
License: This is an open access article distributed under the terms of the Creative Commons Attribution License, which permits unrestricted use, distribution, reproduction and adaptation in any medium and for any purpose provided that it is properly attributed. For attribution, the original author(s), title, publication source (PeerJ) and either DOI or URL of the article must be cited.
License URL: https://creativecommons.org/licenses/by/4.0/

Keywords: Shiguaignathus, Akidnognathidae, Therocephalia, Upper Permian, Naobaogou Formation

Funding: National Basic Research Foundation of China 2012CB821902 NSFC grants 41572019 China-South African Research Project CS08-L02 National Research Foundation of South Africa Chinese Academy of Sciences President’s International Fellowship Initiative Grant 2016VBB054 Jun Liu was supported by National Basic Research Foundation of China (973 grant no. 2012CB821902), NSFC grants (41572019), China-South African Research Project (CS08-L02). Fernando Abdala’s research was supported by the National Research Foundation of South Africa, Conicet from Argentina, and a research trip to China funded by the Chinese Academy of Sciences President’s International Fellowship Initiative Grant (2016VBB054). The funders had no role in study design, data collection and analysis, decision to publish, or preparation of the manuscript.

==============================
The Permian from China has a well-known terrestrial record where approximately 30 tetrapod taxa, including several therapsids, have been described. However, the record of therocephalians in China has remained elusive. Shiguaignathus wangi gen. et sp. nov., discovered in the Member III of the Naobaogou Formation, Nei Mongol, China, is here described. This is the first therocephalian recovered from this fauna and only the second from the Permian of China. It is represented by a well-preserved robust snout of a medium-sized animal. This is the first akidnognathid reported from the Chinese Permian and only the second genus from Laurasia as one genus is known from Russia whereas the remaining members of the group are from the South African Karoo Basin. A phylogenetic analysis of therocephalians supports a basal position of S. wangi within Akidnognathidae, followed by the Russian Annatherapsidus. Akidnognathidae is the latest major group of therocephalian appearing in the fossil record, and one of the few that does not have species from South Africa representing its most basal members.

Introduction

Therocephalians were important components of middle to late Permian terrestrial faunas in Russia and Africa (Abdala et al., 2014; Huttenlocker, Sidor & Angielczyk, 2015; Ivakhnenko, 2011). They represent a rather diverse lineage, especially in the late Permian, with some members, e.g., Glanosuchus in the middle Permian and Theriognathus at the end of the Permian, also being abundant. Two lineages are typical of the middle Permian, namely lycosuchids and scylacosaurids, and they are represented by less than ten species. Three large therocephalian lineages are recognized in the late Permian, Akidnognathidae, Whaitsoidea and Baurioidea, the first two being principally late Permian, while the latter is also well represented in the Triassic (Huttenlocker, 2014).

In China, late Permian tetrapods have been collected from Wutonggou and Guodikeng formations of Xinjiang, the Naobaogou Formation of Nei Mongol, the Shangshihezi Formation of Henan, and the Sunjiagou Formation of Shanxi (Li, Wu & Zhang, 2008). Therocephalians, however, are almost unknown in the Chinese terrestrial Permian. The baurioid therocephalian Urumchia lii was initially reported as late Permian in age (Young, 1952), but subsequently has been demonstrated to be from the Lower Triassic Jiucaiyuan Formation (Sun, 1991). Recently, Liu & Abdala (2017) reported a therocephalian from the Guodikeng Formation at the Dalongkou section of Xinjiang that is most likely latest Permian in age. Other Chinese therocephalians were collected from Triassic layers and are member of the baurioids.

The Naobaogou Formation is known from Nei Mongol and only has a limited distribution within the Daqing Mountains. It conformably overlies the dark purple clastic Shiyewan Formation, and underlies coarse sandstones of the Laowopu Formation. The Naobaogou Formation includes three sedimentary cycles, corresponding to three lithological members (I, II and III), which begin with a thick conglomerate layer and are dominated by purple siltstone. Tetrapod fossils from this Formation were first discovered in 1982 by the team of Yeh H-K, Gao Ke-Qin and Zhu Yang-Long from the Institute of Vertebrate Paleontology and Paleoanthropology of the Chinese Academy of Sciences. This formation produced the dicynodont Daqingshanodon limbus (Zhu, 1989), the captorhinid Gansurhinus qingtoushanensis (Li & Cheng, 1997; Reisz et al., 2011), and tetrapod burrows (Liu & Li, 2013) (Fig. 1). In three field seasons (2009–2011), a team lead by the senior author discovered fossils in the Naobaogou Formation of the Daqingshan area in Nei Mongol. More than 20 dicynodont specimens, five therocephalians and two parareptiles were collected. Most of the specimens come from the middle portion of the Naobaogou Formation, but this probably reflects an uneven collecting strategy, as the vertebrate fossils are known to be distributed throughout the Naobaogou Formation. Considering the fossils recovered, a late Permian age is attributed here to this formation. Therocephalian discoveries confirm the finding of Zhu (1989) who reported members of this group in the Naobaogou Formation, though these fossils are lost and were never described.

Figure 1 (A) Geographical location of the Daqing Mountains fossil localities; (B) simplified geological map of the studied area showing distribution of the Naobaogou Formation and some fossil localities.

Abbreviations: Ar, Archean; C-O, Cambrian-Ordovician; C-P, Carboniferous-Permian; L, Laowopu Formation, N1-3, three members of Naobaogou Formation.

In this paper we describe the first therocephalian from the Naobaogou Formation, represented by a well-preserved snout of a medium-sized animal. This discovery documents the first representative of an akidnognathid therocephalian from China. Members of this group are profusely represented in South Africa and known by a basal representative, Annatherapsidus, from the Russian late Permian (Huttenlocker, Sidor & Angielczyk, 2015). This Russian genus is known from two species, abundantly represented by A. petri and the rare A. postum known only by an isolated lower jaw (Ivakhnenko, 2011). In addition, a specimen of this group was recently identified in the Lower Triassic of Antarctica (Huttenlocker & Sidor, 2012). Besides being the second Chinese Permian therocephalian, this discovery represents only the second akidnognathid genus known from Laurasia.

Nomenclatural acts—The electronic version of this article in Portable Document Format (PDF) will represent a published work according to the International Commission on Zoological Nomenclature (ICZN), and hence the new names contained in the electronic version are effectively published under that Code from the electronic edition alone. This published work and the nomenclatural acts it contains have been registered in ZooBank, the online registration system for the ICZN. A ZooBank LSID (Life Science Identifier) can be resolved and the associated information viewed through any standard web browser by appending the LSID to the prefix http://zoobank.org/. The LSID for this publication is: urn:lsid:zoobank.org:pub: 960FC15A-B1D5-402A-BB16-2FE79A5F0321. The online version of this work is archived and available from the following digital repositories: PeerJ, PubMed Central and CLOCKSS.

Systematic Paleontology

THERAPSIDA Broom, 1905	
THEROCEPHALIA Broom, 1903	
EUTHEROCEPHALIA Hopson & Barghusen, 1986	
AKIDNOGNATHIDAE Nopsca, 1928	
SHIGUAIGNATHUS WANGI gen. et. sp. nov.	

Etymology—‘Shiguai’ refers to the district in which the fossil was collected, ‘gnathus’ is the Greek word for jaw; ‘Wang’ is after Wang Yu, the technician who discovered the specimen.

Holotype—IVPP V 23297, a partial snout lacking its roof.

Type Locality and Horizon—Locality DQS 28, Naobaogou, Shiguai, Baotou, Nei Mongol, China; lower part of Member III, Naobaogou Formation (Fig. 1).

Diagnosis—A medium sized therocephalian showing the following combination of characters: as typical of akidnognathids, present a robust snout, a wide anterior portion of the vomer contacting the premaxilla and five upper incisors. As in Annatherapsidus petri, two processes of the anterior margin of the palatine extend far anterior to the maxilla-palatine foramen. As in some specimens of Promoschorhynchus and of Olivierosuchus, there is only one precanine. Autapomorphies of the new species are the nearly straight and parasagittal crista choanalis directed towards the middle of the suborbital vacuity and ending anterior to the vacuity, the presence of eight postcanines (the largest number reported for an akidnognathid), posterior margin of the choana extending to the level of the third postcanine, and small suborbital vacuity in relation to others akidnognathids.

Description

The specimen is represented by the snout, partially preserved dorsally, but with a pristine palate (Figs. 2 and 3). Taphonomically it is tilted slightly to the left.

Figure 2 Holotype of Shiguaignathus wangi (IVPP V 23297) from the Naobaogou Formation of China.

(A) photograph and (B) line drawing in left lateral view; (C) photograph and (D) line drawing in right lateral view. Abbreviations: J, jugal; L, lacrimal; M, maxilla; PF, prefrontal; PM, premaxilla; SM, septomaxilla; smf, septomaxillary foramen. Photo credit: Jun Liu. Drawing credit: Jun Liu and Yong Xu.

Figure 3 Holotype of Shiguaignathus wangi (IVPP V 23297) from the Naobaogou Formation of China.

(A) photograph and (B) line drawing in dorsal view; (C) photograph and (D) line drawing in ventral view. Abbreviations: ch, choana; EC, ectopterygoid; J, jugal; lo, lacrimal opening; M, maxilla; mpf, maxilla-palatine foramen; pcmt, precanine maxillary tooth; PL, palatine; PM, premaxilla; pmf, premaxillary foramen; PT, pterygoid; SM, septomaxilla; sov, suborbital vacuity; V, vomer. Photo credit: Jun Liu. Drawing credit: Jun Liu and Yong Xu.

The premaxilla forms the anterior end of the snout which projects anteriorly to the level of the first incisors. The midline premaxillary suture is visible in anterior view, but not on the palate. The posterior margin of the premaxilla is in contact with the maxilla and covered dorsolaterally by the septomaxilla (Fig. 2). The suture between the premaxilla and maxilla extends anterodorsally in front to the canine alveolus. The premaxilla forms a ventral plate that encompasses the alveoli of the five upper incisors (Fig. 3). In lateral view, the alveolar margin is nearly horizontal. On the palate, the premaxilla meets the vomer with a short triangular posteriorly-directed vomerine process (Fig. 3B). The premaxillary foramen lies posterior to the alveolus of the first incisor, near the medial suture between the premaxillae, but it can only be observed on the dorsal surface of the bone (Fig. 3A).

A portion of the left dorsal process of the septomaxilla is preserved, whereas the large right septomaxilla is nearly complete with a convex lateral surface well exposed outside of the external naris (Fig. 2B). A small septomaxillary foramen is bordered anteriorly by the septomaxilla and posteriorly by the maxilla.

On the left side, anterior to the canine, the maxilla overlaps the lateral side of the premaxilla at the level of the last incisor. The facial plate of the maxilla is high (more than half of the anteroposterior length), even though its dorsal portion is incomplete (Fig. 2). The maxilla contacts the prefrontal dorsally on the left side, and the lacrimals, partially preserved on both sides, posterodorsally. The lacrimal forms the anterior wall of the orbit, with a small opening for the lacrimal duct. Medially, this bone has a long anterior extension as a thin lamina, which does not reach the antrum Highmori. The lacrimal extends posteroventrally as a long flange and meets the jugal below the orbit. The anterior extension of the jugal is posterior to the anterior margin of the orbit (Fig. 2B).

On the medial side of the maxilla, the canine boss is quite robust as in lycosuchids (Van den Heever, 1994: fig. 5) but in contrast to Moschowhaitsia (Ivakhnenko, 2011, fig. 18a). Anterior to the canine boss, there is a small triangular fossa located posterior to the premaxilla, which is separated by a small horizontal crest from the fossa of the lower canine. This fossa, which is incompletely preserved in the specimen, is called the anterior maxillary fossa (Van den Heever, 1994) or antrum Highmori (Ivakhnenko, 2011) (Table 1; Fig. 4). Ivakhnenko (2011, fig. 18a) identified the medial side of the antrum as partial formed by the premaxilla in Moschowhaitsia, whereas the whole wall is identified as the septomaxilla in Glanosuchus (Hillenius, 1994). Above, the fossa is partially capped by the medially directed maxillary flange. Posterior to the canine boss, there is a larger trapezoidal fossa, which opens dorsally (Fig. 4). It is commonly known as the anterior maxillary sinus (Sigurdsen, 2006), and is called the sinus Highmori here following Ivakhnenko (2011). The high anterior wall of the sinus Highmori is mainly formed by the canine boss, the medial wall by the palatine, the posterior ridge by the palatine and lacrimal, and the lateral wall by the maxilla and lacrimal. A triangular fossa, the palatine sinus, is located posteriorly to the nearly horizontal lacrimo-palatine ridge, and below the orbit. A fenestra penetrates the lacrimo-palatine ridge, so connecting the sinus Highmori and the palatine sinus. The palatine sinus is formed laterally by the lacrimal and palatine, ventrally by the palatine, and dorsally partially covered by the lamina of lacrimal. This fossa is divided by an anteroposteriorly directed low ridge, its medial portion being deeper than the lateral part. Posteromedially, the ectopterygoid forms a low crest, separating the palatine sinus from the suborbital vacuity (Fig. 4).

Figure 4 Holotype of Shiguaignathus wangi (IVPP V 23297) from the Naobaogou Formation of China.

(A) photograph and (B) line drawing in dorsomedial view. Abbreviations: aH, antrum Highmori; cb, canine boss; EC, ectopterygoid; L, lacrimal; ler, lacrimo-ectopterygoid ridge; lpr, lacrimo-palatine ridge; M, maxilla; PM, premaxilla; sH, sinus Highmori; sov, suborbital vacuity; sp, palatine sinus; V, vomer. Photo credit: Jun Liu. Drawing credit: Jun Liu and Yong Xu.

Table 1 Different names for the fossae on medial side of the skull of therocephalians.

Brink (1961)		Maxillary antrum		
Van den Heever (1994)	Anterior maxillary fossa	Posterior maxillary fossa	Posterior maxillary fossa	
Sigurdsen (2006)		Anterior maxillary sinus	Posterior maxillary sinus	
Tatarinov (1999)		The postcanine part of the maxillary sinus	The palatine part of the maxillary sinus	
Ivakhnenko (2011)	Antrum Highmori	Sinus Highmori	Palatine sinus	

In lateral view, most of the ventral margin of the maxilla is slightly concave anteriorly, in front of the level of the canine alveolus and remarkably convex posteriorly (Fig. 2A). In palatal view, the maxilla is constricted behind the canine. The maxilla houses the alveoli for one precanine, one functional canine, one replacing canine and eight postcanines (Figs. 3B, 5). The alveolar buccal margin is slightly laterally concave in ventral view. The buccal margins are placed more ventrally than the lingual margins of the postcanine alveoli. No diastema is presents behind the canine. The posterior accessory canine alveolus is much smaller than the anterior one. The only preserved teeth are the third postcanines on both sides, but they are broken at their bases. These teeth are circular to slightly ellipsoid in palatal view. The maxilla has a broad exposure medial to the canine alveoli, approaching the wide vomer.

Figure 5 Holotype of Shiguaignathus wangi (IVPP V 23297) from the Naobaogou Formation of China.

(A) right, (B) left, maxillary toothrow in palatal view. Abbreviations: fc, functional canine; rc, replacing canine; 1–8, postcanine 1–8. Photo credit: Jun Liu.

The choana is confluent with the fossa for the lower canine, and it extends anteriorly to the level of fourth incisor and posteriorly to the level of the third postcanine. The choana is bordered laterally by the maxilla and palatine, anteriorly by the premaxilla, medially by the vomer, and posteriorly by the vomer and palatine (Fig. 3).

The vomer is wide anteriorly in order to abut the vomerine process of the premaxilla, but is narrow on its posterior end where it contacts with the pterygoid (Fig. 3). It has a long lateral suture with the palatine. The vomer is unpaired with the anterior portion having a ventrally convex surface. A ventromedian crest is present on the central and posterior portion. The vomer is narrowest on its middle part where it is slightly wider than the ventromedian crest, but reaches its greatest width immediately behind the choana. The general structure of the vomer resembles that of Promoschorhynchus (Mendrez, 1974: fig. 2), but the central portion of the bone is much more robust in the African taxon. On the dorsal side, the anterior part of the vomer has a high midline ridge with a lower lateral one on either side (Fig. 3A). The vomeronasal organ lies in the trough formed by these ridges (Hillenius, 2000). The central ridge extends on the whole length of the choana and it is enclosed by the vomerine lateral expansions behind the choana.

In ventral view, the palatine can be divided into a nearly flat lateral portion and a concave medial portion. A sharp crista choanalis is formed where these two portions meet. The crista starts on the maxilla from the anterior level of the canine alveolus and continues posteriorly to the center of the anterior margin of the suborbital vacuity (Fig. 3). On the lateral portion, the palatine bifurcates as two anterior processes located in front of the maxillo-palatine foramen, a narrow medial and a wide lateral one. This pattern is similar to that of Annatherapsidus petri (Ivakhnenko, 2011: fig. 23a). The maxillo-palatine foramen lies at the level of the anterior margin of the second postcanine, and should connect to the sinus Highmori by a groove within maxilla (Sigurdsen, 2006). The anterior margin of the palatine reaches the posterior border of the additional canine alveolus. Laterally, the suture between the maxilla and palatine extends almost parallel with the lateral margin of the skull on the medial side of the postcanine row (Fig. 3B). Posteriorly, the palatine contacts the ectopterygoid and forms the anterior border of the suborbital vacuity. Posteromedially, the palatine contacts the anterior process of the pterygoid.

Only the anterior process and part of the transverse process of the pterygoid is preserved. A ventromedian ridge is continuous with that of the vomer. The pterygoid ridge becomes higher posteriorly and expands laterally to forms a tuber between the posterior margins of the suborbital vacuities (Fig. 3B). Lateral to this ridge, two parasagittal ridges develop on both sides. There are no palatal teeth on the pterygoid. The suborbital vacuity is relatively small, subtriangular and about as long as it is wide.

A relatively small ectopterygoid overlies the anterolateral portion of the transverse process and is in contact with the pterygoid on its medial side, the palatine anteriorly, and the maxilla laterally. There is a palatal ridge lying laterally to the posterior portion of the crista choanalis (Fig. 3B). In the posterior portion of the palate there is a suborbital foramen located laterally to the suborbital vacuity, limited medially by the ectopterygoid and laterally by the maxilla. The foramen opens in the posterior portion of the palatine sinus.

Discussion

Therocephalia is well represented in the Permo-Triassic of Pangea (Huttenlocker & Sidor, 2016; Kemp, 2005). The oldest records are from the middle Permian of the Karoo Basin, South Africa (Abdala, Rubidge & Van den Heever, 2008; Abdala et al., 2014) and it is during the late Permian of the South African Karoo Basin that they become amazingly diverse (Huttenlocker, 2014). As a result of increased field work in East Africa in recent years, the record of African therocephalians outside of the Karoo Basin has improved (Huttenlocker & Sidor, 2016; Huttenlocker, Sidor & Angielczyk, 2015). In Laurasia, late Permian therocephalians are known from Russia (Ivakhnenko, 2011), where they have a good record, although it is comparatively poor in relation to South Africa. Although middle and late Permian deposits in China have revealed a diverse, species-rich vertebrate fauna (Li, Wu & Zhang, 2008), the therocephalian records have proven elusive. Thus, Liu & Abdala (2017) only recently described a therocephalian from the Permo-Triassic transition zone, and Shiguaignathus wangi is only the second Chinese Permian therocephalian.

The new taxon is diagnosed as a therocephalian by the following features: confluence of the palatal fenestra with the internal naris to accommodate the lower canine, anterior expansion of the portion of the vomer separating the choanae which is widest at its contact with the premaxilla, presence of suborbital vacuities bounded by the palatine, pterygoid and ectopterygoid and presence of a ventromedian crest anterior to the interpterygoid vacuity. S. wangi can be further referred to eutherocephalians due to the premaxillary rostral process overhanging incisors and the presence of a fused vomer.

Shiguaignathus wangi presents eight postcanines, the maximum for any akidnognathid (Fig. 5). These teeth are seven in Akidnognathus (Haughton, 1918; Brink, 1961) and less than seven in other akidnognathids. The snout of S. wangi shows a nearly straight and parasagittally directed crista choanalis which is directed towards the middle of the suborbital vacuity and ends anterior to the vacuity, the posterior margin of the choana extends till the level of the third postcanine (Figs. 3 and 6), and the anterior margin of the palatine extends far anteriorly to the maxilla-palatine foramen as two processes. In some therocephalians, such as Moschorhinus kitchingi and Promoschorhynchus platyrhinus, the major part of the crista choanalis is directed to the center of the anterior margin of the suborbital vacuity (Durand, 1991; Mendrez, 1974) (Fig. 6). In these two taxa, however, the posterior part of the crista choanalis is curved posterolaterally and directed towards the lateral side of the suborbital vacuity. The extension of the posterior margin of the choana in other akidnognathids is always beyond the third postcanine (Fig. 6). The palatine bifurcates as two processes extending far anteriorly in front of the maxillo-palatine foramen in Annatherapsidus petri and adults of Theriognathus microps (Huttenlocker & Abdala, 2016; Ivakhnenko, 2011). In most eutherocephalians, the anterior extension of the palatine is only at the level of the maxillo-palatine foramen. It extends anteriorly to the foramen laterally in Pristerognathus polyodon and medially in Ichibengops munyamadziensis (Huttenlocker, Sidor & Angielczyk, 2015; Mendrez, 1975). The maxillo-palatine foramen is located approximately at the level of the first postcanine, slightly anterior to the posterior margin of the choana. The foramen is also close to the posterior margin of the choana in Theriognathus, Ichibengops, Lycideops, Bauria and Scaloposaurus (Huttenlocker & Abdala, 2016; Huttenlocker, Sidor & Angielczyk, 2015; Mendrez, 1975).

Figure 6 Comparison of akidnognathid palates.

(A) Promoschorhynchus (RC 116); (B) Moschorhinus (NHMUK R5698); (C) Annatherapsidus (PIN 2005/1992); (D) Olivierosuchus (NMQR 3605); (E) Shiguaignathus (IVPP V 23297); (F) Akidognathus (SAM-PK-4021). Photo credit: Fernando Abdala and Jun Liu.

Shiguaignathus wangi shares the following features with Akidnognathidae: the septomaxilla enlarged and well exposed outside the external naris, broadly overlapping the premaxilla anteriorly; the anterior margin of the vomer greatly expanded; the jugal anterior extent restricted to the anterior margin of the orbit; and the presence of five upper incisors and functional precanine teeth. Different from all other akidnognathids except for Olivierosuchus, S. wangi features a maxillary postcanine alveolar margin which is slightly concave laterally in ventral view.

Phylogenetic analysis

The new species was coded following Huttenlocker & Sidor’s (2016) data matrix with redefinition of some characters and revision of two codings (Appendices I, II) and analyzed with TNT 1.5 (Goloboff & Catalano, 2016). Three outgroups, Anomodontia, Biarmosuchus and Titanophoneus, were discarded from the final data matrix because Gorgonopsia is generally accepted as the sister group of Therocephalia plus Cynodontia (Rubidge & Sidor, 2001) and exploration of the data by phylogenetic analyses showed that there was no change in the placement of the groups of interest for this study. The routine followed for the search of most parsimonious trees (mpt) consisted of 10 random addition sequences and TBR, saving 10 trees per replications, and a second search using the trees from RAM as starting point and implementing TBR on those trees. The search resulted in 3,700 mpt of 368 steps in which major groups of Therocephalia, excepting Scylacosauridae, are monophyletic in the strict consensus (Fig. 7 left). S. wangi is placed either within Akidnognathidae or as the sister taxon of other akidnognathids. The majority rule consensus tree (Fig. 7 right) is nearly identical to the strict consensus represented in the fig. 5b of Huttenlocker & Sidor (2016) but relationships among Bauroidea is most similar to their Bayesian consensus, represented in their fig. 5a, by the recovery of a monophyletic Lycideopidae. Bremer support of 1 and low values of bootstrapping characterize most of the major lineages of therocephalians, being Bauridae the only exception with Bremer support of 2 and resampling values above 50.

Figure 7 Strict (left) and majority (right) consensus trees of therocephalians.

The numbers on the right side indicate the frequency of clades in the fundamental trees.

There are three characters supporting the sister group relationship of Akidnognathidae and Chthonosauridae: vomer completely fused (character 43), posterior apophysis of epipterygoid contacting the prootic (character 57), and the absence of diastema behind dominant caniniform (character 127), but these characters are only known in one taxon of the Chthonosauridae. Two of these characters (43 and 127) supported the inclusion of S. wangi within this clade in some of the mpt. The synapomorphy of Akidnognathidae is the presence of vomer greatly expanded anteriorly with the anterior width around half of the vomer length between choanae (character 40).

The majority consensus tree (Fig. 7 right) shows the basal placement of the Chinese S. wangi and the Russian A. petri, suggesting a possible origin of Akidnognathidae in Laurasia. This is the only major group of Therocephalia in which basal representatives are not from South Africa. Other than S. wangi whose age is unsure right now, the oldest akidnognathids are known from the Cistecephalus AZ of the Karro Basin, represented by Akidnognathus and the bizarre Euchambersia (Huttenlocker et al., 2011). The youngest history of the group is then developed only in South Africa. Akidnognathids is the last lineage of Therocephalia to appear in the fossil record as representatives of the Baurioidea and Whaitsioidea are known from levels of the Pristerognathus AZ (Huttenlocker, 2014). Two akidnognathid taxa from the Karoo Basin, the abundant Moschorhinus and the scarce Promoschorhynchus, cross the P/T boundary and the last survivor is Olivierosuchus, a taxon which is only modestly represented in the Lystrosaurus AZ.

Conclusion

The discovery of S. wangi, the first therocephalian from the Naobaogou Formation, increases the diversity of this fauna, and reinforces the presence of akidnognathid therocephalians in Laurasia. The phylogenetic analysis places the Laurasian species as basal members of this group of therocephalians, and suggests the putative origin of akidnognathid in Laurasia, the only case for a major group of therocephalians.

Supplemental Information

Supplemental Information 1 The character list and data matrix used in this paper

Click here for additional data file.

Supplemental Information 2 The data matrix in tnt format

Click here for additional data file.

We thank the field team that went to Daqingshan in 2009 (Wang Yu, Jia Zhen-Yan, and Chang Shao-Ning) for its hard work, and especially Wang Yu who collected the specimen, Fu Hua-Lin who prepared it, and Xu Yong who made the drawing. Reviews by Adam Huttenlocker and Christian Kammerer and revision of the English by Cynthia Kemp are especially appreciated.

Institutional abbreviations

BP Evolutionary Studies Institute, University of the Witwatersrand, Johannesburg, South Africa

IVPP Institute of Vertebrate Paleontology and Paleoanthropology, Chinese Academy of Sciences, Beijing, China

NMQR National Museum, Bloemfontein, South Africa

NHMUK Natural History Museum, London, UK

PIN Paleontological Institute, Russian Academy of Sciencies, Moscow, Russian

RC Rubidge collection, Wellwood, Graaff-Reinet, South Africa

SAM Iziko Museums-South African Museum, Cape Town, South Africa

USNM National Museum of Natural History, Washington D.C., USA

Additional Information and Declarations

Competing Interests

Author Contributions

Data Availability

New Species Registration

The authors declare there are no competing interests.

Jun Liu conceived and designed the experiments, performed the experiments, analyzed the data, contributed reagents/materials/analysis tools, wrote the paper, prepared figures and/or tables, reviewed drafts of the paper.

Fernando Abdala performed the experiments, analyzed the data, contributed reagents/materials/analysis tools, wrote the paper, reviewed drafts of the paper.

The following information was supplied regarding data availability:

The specimen is housed in the Institute of Vertebrate Paleontology and Paleoanthropology, Chinese Academy of Sciences as IVPP V 23297. The character list and the data matrix used in this article can be found in the Supplemental Information.

The following information was supplied regarding the registration of a newly described species:

Publication LSID: urn:lsid:zoobank.org:pub:960FC15A-B1D5-402A-BB16-2FE79A5F0321

Shiguaignathus: urn:lsid:zoobank.org:act:E24B7100-D4DB-4C0C-90FD-098065209EC3

Shiguaignathus wangi: urn:lsid:zoobank.org:act:C55A587B-8A34-407E-AD8B-C72609075D4F.

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
