# Peer review of "The tetrapod fauna of the upper Permian Naobaogou Formation of China: 1. Shiguaignathus wangi gen. et sp. nov., the first akidnognathid therocephalian from China"

_PeerJ, doi:10.7717/peerj.4150_

## Round 0.1 · original submission · Minor Revisions

Please address the reviewers' comments in detail. The manuscript must also be edited by a fluent English speaker before it can be considered for publication. There are many minor mistakes, especially missing articles. As PeerJ does not provide copy-editing this is the authors' responsibility.

·

Basic reporting

The manuscript has been improved since last I reviewed it, and my comments are fewer, see below. I remain unconvinced that this is not just a Chinese specimen of Annatherapsidus, but at this point the authors and I will just have to agree to disagree on this point. In particular, though, I have doubts as to the presence of eight upper postcanines in this specimen. On the right maxilla there are only five alveoli, and that large of a discrepancy in tooth count between sides would be highly unusual in a therocephalian. It is difficult to see the alveoli in the figure provided, and it appears the alveoli in the left maxilla may not be fully prepared. A better figure of the specimen in palatal view would greatly improve this situation and make the identification as a new taxon much more convincing.

Line 12: “well know” should be “well-known”
Line 18: “akidnognathid” is misspelled
Line 19: “one species is known from Russia”—incorrect. There are two species of Annatherapsidus from Russia (A. petri and A. postum). (Furthermore, there are other Russian therocephalians that may be akidnognathids based on some phylogenetic analyses, e.g., Scylacosuchus.)
Line 30: “component” should be “components”
Line 35: “represented by a few species”—well, there are 55 named species of these therocephalians, none of which have been formally synonymized outside of an unpublished PhD thesis, so it is hard to say. I agree that once all the revisionary work is done there will be few left, however.
Line 68: As noted above, multiple species of Annatherapsidus from Russia are known. See Ivakhnenko (2011).
Line 87: “von Nopsca” should just be “Nopcsa” (note spelling)
Line 170: “is much robust” should be “is much more robust”
Lines 203–204: “Middle Permian” and “Late Permian” are no longer formal chronostratigraphic subdivisions, please use the informal “middle Permian” and “late Permian” instead.
Lines 209–211: Change this sentence to “Although middle and late Permian deposits in China have revealed a diverse, species-rich vertebrate fauna, therocephalian records have proven elusive.”
Line 264: Throughout this paragraph the family names Akidnognathidae and Perplexisauridae need to be capitalized. Also, why is it called Perplexisauridae when Figure 5 (and existing therocephalian literature) calls this clade Chthonosauridae?
Line 288: “akidnognathid” should be “akidnognathids”
Figure 1: It would be easier to interpret this figure if you note the type locality of Shiguaignathus by listing the genus next to the dot, like you do for Gansurhinus and Daqingshanodon.
Figure 5: “Euchambersia” and “Lycideopidae” are misspelled. Also, the genera “Ophidostoma” and “Microwhaitsia” are currently unpublished. Unless you are absolutely certain that these names will be published before the Shiguaignathus manuscript, please replace these names with the respective specimen numbers so as not to create nomina nuda.

Experimental design

See above.

Validity of the findings

See above.

·

Basic reporting

Liu & Abdala present a well-preserved palate and snout of a putative new taxon of akidnognathid therocephalian. The overall presentation and illustrations are good, although they would benefit greatly from comparative photos of other akidnognathids and close-up illustrations of some of the purported anatomy, especially the dentition (or what little of the dentition is preserved).

Experimental design

The experimental design is good and follows standard geological, paleontological, and phylogenetic procedures.

Validity of the findings

The findings are important and convincing with regards to the presence of an akidnognathid therocephalian in this fauna. However, I have some questions about the generic diagnosis and justification of the new taxon (see detailed comments below).

Additional comments

I congratulate the authors on their discovery of a very nice centerpiece specimen for this study, and on the excellent presentation of their paper. Regarding my specific concerns, I want to make clear to the authors & editors that I have reviewed a previous version of this manuscript that was submitted to another journal. At the time, I did not feel that it merited publication in its current form because the major claim that the authors were erecting a new taxon was not justified by the available information in the fossil as illustrated. Since then, there's been some effort to address my concerns, although the authors continue to treat the specimen with a new genus name. I take issue with most of the characters in their diagnosis, but recognize that one or two subtle proportional traits could be useful in erecting a new taxon, and I am open-minded about these--namely the possibility of a high postcanine count and the obviously small size of the suborbital vacuity.

Taking on the the 'Diagnosis' point-by-point, the shape of the choana looks identical to the Russian Annatherapsidus and other early akidnognathids (it is not autapomorphic as their Diagnosis suggests), the extra palatal ridges are also present in Olivierosuchus and Promoschrhynchus (and perhaps others), and the postcanine tooth number appears at first consistent with the primitive number (6-7) based on the photographs, although they dashed-in extra putative alveoli in their drawing that suggest as many as eight postcanine positions--note that a previous version of the manuscript presented only 6 on one side and 7 on the other, which would be consistent with Akidnognathus. Given the ambiguity, I'd like to see this better illustrated with a close-up photo before I am convinced of this, but I suppose it's possible there may be up to eight.

The most convincing evidence of a new taxon is in the extremely small suborbital vacuities. I agree with the authors in that I have not seen vacuities this small in another akidnognathid specimen, so perhaps the authors are on to something here. Maybe the importance of this structure should be emphasized for now as evidence of a new taxon until more complete specimens are found in the future?

Overall, the new record adds important information about the biogeography and biochronology of Permian therocephalians, and I think it should be published in some form. I will make the recommendation to publish with minor revisions, pending that the authors can (1) add one or two useful comparative photos from the palates of other specimens, (2) better illustrate the dentition to support their claim about the high number of postcanines, which is not super convincing right now, and (3) revise the Diagnosis accordingly. There are also a number of minor spelling/grammar errors in the manuscript and figures (e.g., Lycideopidae is misspelled and placed at the wrong node in the cladogram) that the authors/editors will need to address at some point, as I am only commenting on the manuscript's scientific value at this point.

Thank you and good luck!

Adam Huttenlocker
University of Southern California

---

## Round 0.2 · Minor Revisions

Please add the references for the taxonomic categories listed in lines 81-84 at the beginning of the section on systematic paleontology. All references with authors and dates must be matched by bibliographic citations.

---

## Round 0.3 · accepted · Accept

The revised version is recommended for acceptance for publication.